Ensemble graph auto-encoders for clustering and link prediction

Xie Chengxin 1 2
Huang Jingui 2
Shi Yongjiang 1
Pang Hui 1
http://orcid.org/0009-0003-9723-5382 Gao Liting 1 gaoliting@163.com
Wen Xiumei 1 xiumeiwen@163.com
1 Hebei University of Architecture , Zhangjiakou , China
2 Hunan Normal University , ChangSha , China
Piangerelli Marco
Electronic publication date: 2025 Jan 22
Publication date: 2025
Volume: 11
Electronic Location ID: e2648
Received 2024 Jul 8; Accepted 2024 Dec 16
Copyright: © 2025 Xie et al.
Copyright year: 2025
Copyright holder: Xie et al.
License: This is an open access article distributed under the terms of the Creative Commons Attribution License, which permits unrestricted use, distribution, reproduction and adaptation in any medium and for any purpose provided that it is properly attributed. For attribution, the original author(s), title, publication source (PeerJ Computer Science) and either DOI or URL of the article must be cited.
License URL: https://creativecommons.org/licenses/by/4.0/

Keywords: Graph auto-encoders, Low embedding, Ensemble, Link prediction, Clustering

Funding: Science and Technology R&D Platform Construction Special Project 225676106H Hebei Energy Storage and Heat Technology Innovation Centre This work is based on the research project Science and Technology R&D Platform Construction Special Project [225676106H]—Performance Grant for Hebei Energy Storage and Heat Technology Innovation Centre. The funders had no role in study design, data collection and analysis, decision to publish, or preparation of the manuscript.

==============================
Graph auto-encoders are a crucial research area within graph neural networks, commonly employed for generating graph embeddings while minimizing errors in unsupervised learning. Traditional graph auto-encoders focus on reconstructing minimal graph data loss to encode neighborhood information for each node, yielding node embedding representations. However, existing graph auto-encoder models often overlook node representations and fail to capture contextual node information within the graph data, resulting in poor embedding effects. Accordingly, this study proposes the ensemble graph auto-encoders (E-GAE) model. It utilizes the ensemble random walk graph auto-encoder, the random walk graph auto-encoder of the ensemble network, and the graph attention auto-encoder to generate three node embedding matrices Z. Then, these techniques are combined using adaptive weights to reconstruct a new node embedding matrix. This method addresses the problem of low-quality embeddings. The model’s performance is evaluated using three publicly available datasets (Cora, Citeseer, and PubMed), indicating its effectiveness through multiple experiments. It achieves up to a 2.0% improvement in the link prediction task and a 9.4% enhancement in the clustering task. Our code for this work can be found at https://github.com/xcgydfjjjderg/graphautoencoder.

Introduction

Graph structures play a pivotal role in complex data networks, such as traffic networks, protein interaction networks, and paper citation relationships (Pei et al., 2021; Ma, Na & Wang, 2021; Grover & Leskovec, 2016; Huang, Silva & Singh, 2021; Tang, Yang & Li, 2022; Kipf & Welling, 2017; Pan et al., 2019). A convolutional neural network is an essential model for deep learning, in which an AE uses a CNN as an encoder to extract feature information (Han & Ghadimi, 2022). Although autoencoders are classic models in deep learning and excel in unsupervised learning scenarios, they primarily rely on stacking nonlinear layers and utilizing nonlinear transformations to capture intricate relationships within input features (Tang, Yang & Li, 2022; Kipf & Welling, 2016; Xie et al., 2022a). However, when dealing with irregular graph data in the real world, traditional autoencoders fail to harness the inherent structural information within the graph (Cheng et al., 2024). Kipf & Welling (2016) proposed the graph autoencoder to address this issue, allowing the utilization of graph structural data and achieving node embedding by encoding the adjacency matrix and eigenmatrix of nodes, reconstructing the original adjacency matrix through inner product operations (Kipf & Welling, 2016; Weng, Zhang & Dou, 2020; Kipf & Welling, 2017; Xie et al., 2022a; Wou et al., 2021).

Graph embedding, a fundamental area within graph neural networks, focuses on converting high-dimensional vector matrices into lower-dimensional vector representations through mapping techniques (Huang, Silva & Singh, 2021; Yang et al., 2015; Wou et al., 2021; Li et al., 2022; Xia et al., 2023). Therefore, graph auto-encoders have gained prominence. Despite their simplicity and effectiveness in tasks such as clustering and link prediction, existing graph auto-encoder models tend to disregard the embedding representations of nodes, resulting in poor embedding effects (Weng, Zhang & Dou, 2020; Huang & Frederking, 2019). Unlike traditional models such as TADW (Yang et al., 2015), DNGR (Cao, Lu & Xu, 2016), and GraRep (Cao, Lu & Xu, 2015), which predominantly use either structural or node data from the graph, graph auto-encoders employ graph convolutional neural networks as encoders (Kipf & Welling, 2016; Zhang, Wang & Liu, 2021). This enables better node representation by continually aggregating information from neighboring node features (Cheng et al., 2024; Chen et al., 2023; Wu et al., 2023). However, graph data usually contain nodes that are not directly connected but are structurally similar. Existing graph auto-encoders fall short of capturing the information of these contextual nodes, posing a new challenge (Huang & Frederking, 2019; Lin Li & Liu, 2021).

Compared to models such as DeepWalk (Perozzi, Al-Rfou & Skiena, 2014), Node2vec (Grover & Leskovec, 2016), and DDRW (Li, Zhu & Zhang, 2016), which exclusively consider graph structure for modeling, graph auto-encoders utilize graph structure and node features for modeling to achieve better node representation. Pan et al. (2019) proposed a graph auto-encoder with adversarial regularization to obtain better node embedding representations. This model addresses the poor embedding quality caused by traditional graph auto-encoders neglecting latent data distribution within the graph. However, it overlooks the contextual information of the nodes. Unlike graph auto-encoders focusing on graph structural information, Hou et al. (2023) proposed the GraphMAE2 model, incorporating a multi-view random re-masking strategy and an enhanced decoder to learn graph representations for model feature reconstruction. However, GraphMAE2 prefers local structures and does not fully consider the capture of graph structural information and node contextual information. These models all employ a single encoder structure, which presents limitations in node representation.

There are significant limitations, unlike the graph embedding models introduced earlier, which only use a single model to extract node features and higher-order association information from the graph (Qiaoyu, Ninghao & Hu, 2023). Although recent work has attempted to use different strategies (such as masking techniques) to obtain more node features, these techniques do not address the inability of a single model to capture all the information embedded in the graph. The ensemble embedding method is proposed in response to the shortcomings of existing graph embedding models, which are not considered by most existing methods. This method effectively addresses the inability of existing models to capture contextual node information and obtain more node features to better mine graph patterns.

This study designs a novel ensemble model called the ensemble graph auto-encoder (E-GAE). A new encoder is reconstructed based on the multi-view integration concept to capture the feature representation of nodes. E-GAE addresses the issues found in existing models, such as the neglect of node embedding representation during reconstruction and the failure to capture contextual node information, resulting in low-quality embeddings. The contributions of this study can be summarized as follows:

1. The ensemble model E-GAE is proposed based on the ensemble learning concept. It was discovered that the ensemble of three different models yields the best results by integrating various numbers of graph auto-encoders. This ensemble model significantly enhances the capability of extracting node features.

2. The E-GAE model is introduced to address the low-quality embeddings resulting from existing models that ignore node embedding representation during reconstruction and cannot capture contextual node information. This model independently trains a random walk graph auto-encoder, a graph attention auto-encoder, and an ensemble network random walk graph auto-encoder. The node embedding matrix Z generated by each model is integrated using adaptive weights to reconstruct a new node embedding matrix H, effectively mitigating the problem of low embeddings.

3. The existing graph auto-encoder model is modified to address the limitations of using only a single encoder model to learn the embedding representation of nodes. Extensive proof-of-concept experiments conducted on multiple benchmark datasets indicate that the proposed E-GAE model is superior in link prediction and clustering tasks.

Related work

Nowadays, social networks have come under increasing scrutiny (Xie et al., 2023). The study of social networks encompasses various methodologies, and graph neural networks have garnered significant attention. Among the various directions in graph neural networks, graph autoencoders play a crucial role in graph embedding, clustering, and link prediction. Node embedding, a common technique for representing graph data in lower-dimensional spaces, has been widely used in graph embedding endeavors (Xie et al., 2022a, 2023; Wang et al., 2019; Li et al., 2022; Xia et al., 2023). However, traditional graph encoders, although efficient in reducing graph data for loss reconstruction, tend to ignore node representation, leading to low-quality embedding challenges (Salehi & Davulcu, 2019).

Huang & Frederking (2019) introduced a model known as the restarting random walk graph autoencoder (RWR-GAE) in response to the problems of low embedding and the inability to capture contextual node information. Although the RWR-GAE model uses restart random walks to consider the node context within the graph topology, it does not solve the low embedding problem. The model only uses a single graph convolutional network (GCN) as the encoder, which has many shortcomings, and the experimental results of the model cannot be reproduced. However, this method provides an optimization idea. Traditional autoencoders fall short in using the structural relationships within graph data. Salehi & Davulcu (2019) proposed the graph attention autoencoder (GATE) to enhance node features and graph topology utilization. The GATE model introduces the attention mechanism based on GAE, focusing more on the graph’s topology and the nodes’ correlation. However, this method has challenges in capturing node context information and cannot entirely mine graph patterns, making it difficult to solve the problem of low embedding.

The prevalence of noisy data within graph datasets interferes with node feature extraction, causing poor clustering effects and diminishing the model’s effectiveness and robustness (Xie et al., 2023; Mrabah et al., 2022). Reconstructing the adjacency matrix as part of the modeling process can lead to the learning of irrelevant features, resulting in low-quality node embedding. Mrabah et al. (2022) introduced a clustering method based on a reconstructed GAE, which utilizes sampling operators and operator correction to mitigate some of these issues. However, the GAE model uses GCN as an encoder, and GCN ignores higher-order semantic information and the correlation between nodes when extracting node features. This results in low embedding problems and affects downstream tasks.

Pan et al. (2019) contended that existing graph autoencoders often overlook the underlying data distribution despite the emergence of several recent graph autoencoder models. They proposed the Adversarial Regularized Graph Autoencoder (ARGA) and Adversarial Regularized Variational Graph Autoencoder (ARVGA) models to address the low embedding problem. However, these models only offer partial mitigation. Another limitation is that existing graph autoencoders tend to be shallow models (Pei et al., 2021; Cheng et al., 2024; Xie et al., 2023; Kang et al., 2022). Wu & Cheng (2022) introduced the deep DGAE model to enhance node embedding representations, which uses jump connections to augment node properties. However, the DGAE model does not effectively capture node context information and risks over-smoothing as the number of layers increases. Zhang, Wang & Liu (2021) proposed a graph autoencoder model with node attribute enhancement (NEGAE), which employs graph convolution for feature extraction and graph reconstruction via inner product operations. However, this model still relies on a single encoder approach, posing limitations.

Traditional models tend to employ GCNs as encoders for tasks related to nearest-neighbor graph mining (Wang et al., 2019). Although this approach is effective, it has limitations when the graph structure plays a vital role. Cheng et al. (2024) introduced the NWR graph autoencoder, which reconstructs proximity and topology but still relies on a single model for node representation and does not effectively solve the low embedding problem. Tsitsulin et al. (2023) indicated that while graph neural networks have excelled in graph embedding, they struggle with graph clustering tasks. They introduced a complex network known as DMoN, which, despite its complexity, fails to resolve the node embedding issue effectively. Existing deep clustering algorithms often disregard structural information and essential node generation features, leading to the application of inappropriate clustering methods for embedding. Zhang et al. (2023) proposed the EGAE model, demonstrating that, under certain conditions, the optimal division based on inner product distance can be derived by relaxing the k-means algorithm. However, this model still uses a single encoder and cannot capture the higher-order information in the graph and the correlation between nodes. It cannot effectively mine the graph pattern, so it cannot completely solve the low embedding problem.

Compared to the previous model, Hou et al. (2022) introduced a mask graph autoencoder (GraphMAE) to solve the problem of feature loss in reconstructing traditional graph autoencoders. The masking strategy is proposed to reconstruct node features, but the model does not require reconstructing the graph structure and ignores the context information and higher-order semantic information of nodes in the graph.

This study proposes a novel graph embedding method, E-GAE, which uses a random walk strategy to obtain node context information. The model integrates three encoder models and is reconstructed into a new encoder, which can effectively mine the complex information in the graph pattern and the complex correlation between nodes from multiple angles, obtaining high-quality node representation to provide guarantees for downstream tasks.

Proposed model

Graph convolutional networks

The effective utilization of graph data has become a pressing concern, leading researchers to explore the field of graph neural networks. Kipf & Welling (2017) introduced a novel approach for message aggregation and successfully applied it within the traditional domain of graph neural networks (Kipf & Welling, 2017). Currently, the most widely used graph convolutional neural network (GCN) is the null domain-based GCN (Xie et al., 2022b). The concept of GCN is derived from CNN’s strategy in handling grid data, with the core idea of aggregating node-level information from neighborhoods to more flexibly and effectively obtain the feature representation of nodes (Xie et al., 2023). In traditional GCNs, the process of updating node features H(l+1) can be represented by Eq. (1):

(1) H(l+1)=σ(D~−12A~D~−12H(l))

where Ã = A + I, I is the unit matrix, D~−12 is the normalization operation of the degree matrix D, H(l) is the feature representation of the node at the current layer of nodes by aggregating the features of the surrounding neighboring nodes, σ is the nonlinear activation function, and H(l+1) is the representation of the node at the next layer.

GAE is an unsupervised learning model based on GCN, designed to aggregate critical neighborhood information and mine the topological features of graph structures. However, traditional GCN methods focus only on specific local network structures and pay insufficient attention to high-level information such as information flow and node correlations, which results in poor performance in complex network analysis. Existing methods focus more on capturing and utilizing graph structural information to address this issue. For instance, Qiaoyu, Ninghao & Hu (2023) proposed a rebalanced neighbor discovery strategy, category-enhanced negative sampling methods, and adversarial learning to improve recommendation diversity and address directional issues based on diversity problems in recommendation systems. However, these methods are focused only on domain-specific designs, limiting their generalizability and flexibility. In response to the existing problems, this study designs a universal integrated algorithm model, E-GAE. E-GAE adopts a multi-view approach to expand the model’s receptive field and constructs a new node embedding matrix by integrating three different graph autoencoders to effectively extract critical features. This integrated mechanism is built to enhance the model’s universality in extracting key feature patterns in any topology of graph structures.

Loss function

(2) Loss=−1N∑i=1nyi.log⁡(p(yi))+(1−yi).log⁡(1−p(yi))

where y is the binary label 0 or 1, and p(y) is the probability of output belonging to y labels. Binary cross entropy is employed to evaluate the quality of the prediction results of a binary classification model, i.e., if the predicted value p(y) approaches 1, then the value of the loss function should approach 0 for the case labeled y is 1. In contrast, if the predicted value p(y) approaches 0 at this time, the value of the loss function should be enormous.

This study used the binary cross-entropy loss function for all three models, and the total loss is shown in Eq. (3):

(3) Ltotal=w1∗L1+w2∗L2+w3∗L3

where Ltotal is the total loss, wi is the hyperparameter, and Li is the loss for each part.

Ensemble graph auto-encoder (E-GAE)

Graph attention auto-encoder (GATE)

Traditional graph autoencoders overlook the importance of reconstructed graph structure and node features. Salehi & Davulcu (2019) introduced the GATE model to address this issue. In GATE, the encoder utilizes node attributes to acquire feature representations of the original nodes and utilizes the graph’s topology by employing stacked layers to generate node embedding representations (Weng, Zhang & Dou, 2020; Salehi & Davulcu, 2019; Cheng et al., 2024). The fundamental architecture of the attention-based graph autoencoder employs multiple encoder layers, mainly because increasing the number of encoder layers enables a deeper model hierarchy, enhancing the model’s capacity to learn and yield superior node representations (Salehi & Davulcu, 2019; Xie et al., 2022a). In addition, it allows for the comprehensive utilization of graph topology data to improve node embedding representations. Each encoder layer aggregates information from neighboring nodes based on the node neighborhood to create an embedding representation of the node (Salehi & Davulcu, 2019). The model addresses the challenge of determining attention coefficients between nodes and their neighbors by incorporating an attention mechanism with shared parameters among nodes within the same layer. The specific approach is illustrated in Eq. (5):

(4) ei,j(k)=Sigmoid(vm(k)Tσ(w(k)Hi(K′))+vn(k)Tσ(w(k)Hj(K′)))

where w(k), Vm(k), and Vn(k) are the parameters trainable at the kth encoder layer, Hi(k′) is the node representation of node i reconstructed at layer k, Hj(k′) is the node representation of node j reconstructed at layer k, σ is the activation function, and Sigmoid is the activation function. ei,j(k) is the attention coefficient between neighboring node j and node i in the kth decoder layer.

The decoder uses the same number of layers as the encoder. Each decoder layer requires solving the corresponding encoder layer in reverse. In other words, each decoder layer reconstructs a node’s embedded representation based on its neighbors’ representation around the node.

Random walk graph auto-encoder (RWR-GAE)

Huang, Silva & Singh (2022) reported that existing graph autoencoder models predominantly emphasize reconstruction loss and graph structure but overlook the data distribution of potential codes in the graph, leading to a low node embedding effect. They introduced the ARGA model. However, Huang & Frederking (2019) indicated that this approach is not ideal, prompting them to propose the RWR-GAE model. Algorithm 1 describes the specific process of RWR-GAE.

Algorithm 1 Random Walk Graph Auto-encoders.

Input: Graph G(V, X, A), window size ω, walks per epoch t, walk length l, restart probability α	
Z ← Encoder(G)	
V’ = shuffle(V )	
O = sample t vertices from V’	
  for each νi ∈ O do	
     Wνi = Random_Walk(A, νi, l, alpha)	
     for each νj ∈ Wνi do	
       for each µk ∈ Wνi [j − w : j + w] do	
        Lνj=−log⁡Pr(μk|Z(vj))	
       Update SkipGram and Encoder using ▽ Lνj	
A^ ← Decoder(Z)	
Update Decoder and Enoder	
END	

The random walk algorithm commences by initiating traversal from a vertex or a series of vertices within the graph. During the traversal, all nodes are traversed with a probability of p to other nodes in the graph, while with a probability of 1-p, the walk proceeds to neighboring nodes of the current node (Huang, Silva & Singh, 2022; Hoskins et al., 2020). After each walk is completed, a probability distribution is generated, reflecting the likelihood of visiting each node within the graph. This probability distribution is employed as input for the subsequent walk, and the process iterates. This probability distribution converges when specific conditions are met, culminating in a smooth probability distribution (Hoskins et al., 2020). The random walk graph auto-encoder employs random walks to acquire a sequence of traversed nodes. This sequence of nodes is subsequently input into the Skip-gram model (Huang & Frederking, 2019; Huang, Silva & Singh, 2022). The Skip-gram model, a variant of the Word2Vec model, predicts the context of a central word given its context words (Huang & Frederking, 2019; Wou et al., 2021). The Skip-gram model takes n-node features as input proceeds through a hidden layer to derive a hidden layer representation of the nodes and reduces computational complexity through optimization using down-sampling techniques. This process allows for the prediction of context nodes and the generation of a low-dimensional representation of the nodes (Wou et al., 2021).

EGSRWR-GAE

Existing graph auto-encoder models predominantly utilize the GCN as the sole encoder (Cheng et al., 2024; Kollias et al., 2022; Salha et al., 2021). Although these models are simple, they exhibit limitations in feature extraction when a single model is an encoder (Ma, Na & Wang, 2021; Lin Li & Liu, 2021; Li, Zhang & Zhang, 2022; Xie et al., 2023). Current models primarily employ shallow architectures with poorly receptive fields, and GCNs are exclusively used for transductive learning (Wu & Cheng, 2022; Li, Zhang & Zhang, 2022). As a new encoder, the EGSRWR-GAE model employs three distinct networks, GCN, GAT, and SuperGAT, effectively addressing the drawback of sole-model usage. The model structure is illustrated in Fig. 1. EGSRWR-GAE uses these three network models to acquire node representations. These learned representations are integrated through adaptive weighting to generate new node representations. Unlike traditional graph auto-encoders, which rely on a sequential approach to learn node features, EGSRWR-GAE adopts an innovative ensemble approach to learn node feature representations using three different network models. Subsequently, the learned node features are aggregated utilizing weights to reconstruct new node features. The regenerated node representations are then employed to derive the node embedding representation matrix Z. The original graph structure is reconstructed in the decoding phase using the inner product (Tang, Yang & Li, 2022; Huang & Frederking, 2019; Salehi & Davulcu, 2019; Cheng et al., 2024; Wu & Cheng, 2022). Compared to existing models, this method utilizes multiple network models and implements the ensemble concept to address the issue of low embedding caused by the reconstruction losses in current graph auto-encoder models.

Figure 1 The overall architecture of the EGSRWR-GAE model.

The ensemble graph auto-encoder (E-GAE) is introduced to address the limitations of existing graph auto-encoder models, characterized by disregarding node representation in reconstruction losses and the inability to capture contextual information of nodes in graph data, which results in lower embeddings. The model structure is illustrated in Fig. 2. Algorithm 2 describes the specific process of E-GAE.

Figure 2 The overall architecture of the E-GAE model.

Algorithm 2 Ensemble graph auto-encoder (E-GAE).

Input: Graph G(V, X, A), window size ω, walks per epoch t, walk length l, restart probability α	
RWR-GAE ← Graph(A, X w, l, t, a)	
Z1 ← RWR-GAE	
GATE ← Graph(A,X)	
Z2 ← GATE	
EGSRWR-GAE ← Graph(A, X, w, l, t, a)	
Z3 ← EGSRWR-GAE	
Z ← w1 * Z1 + w2 * Z2 + w3 * Z3, wi is adaptive weights	
Â ← Decoder(Z)	
Calculate loss	
END	

The proposed EGSRWR-GAE model reconstructs new node features using three networks, whereas the E-GAE model reconstitutes the node embedding matrix by integrating three types of graph auto-encoders. The E-GAE model combines three graph auto-encoders, with each taking the node’s feature matrix X and adjacency matrix A as inputs to produce node embedding matrices Z1, Z2, and Z3, respectively. The node embedding matrices are then combined using adaptive weights to yield a new node embedding matrix H, as demonstrated in Eq. (5):

(5) H=W1∗Z1+W2∗Z2+W3∗Z3

where H is the new node embedding matrix, Zi is the ith node embedding matrix, W1, W2, and W3 are trainable hyperparameters, and A’ is the reconstructed adjacency matrix.

The proposed E-GAE model is integrated from three graph auto-encoders, but the possibility of integrating more graph auto-encoders is explored. Hence, additional experiments are conducted. The effects of integrating two, three, four, and five models are compared in the experiment. GATE+EGSRWR-GAE is an integration of two models, abbreviated as GE. GATE+EGSRWR-GAE+RWR-GAE+GAE is an integration of four models, abbreviated as GERG. GATE+EGSRWR-GAE+RWR+GAE(TAGconv)+GAE(ARMAconv) integrates five models, abbreviated as GERTA. GAE(TAGconv) denotes replacing the graph auto-encoder GCN layer with TAGconv. GAE(ARMAconv) denotes replacing the graph auto-encoder GCN layer with ARMAconv. The model effects are validated on the Cora and Citeseer datasets. The experimental results are listed in Table 1. Through experimentation, it is evident that integrating more models does not always produce better results. The E-GAE model changes the architecture of the traditional model and solves the disadvantages of using a single model by integrating multiple models. Unlike previous models that rely on a single model to learn the embedded representation of nodes, which has apparent drawbacks, the E-GAE model adopts an ensemble learning approach to overcome the limitations of a single weak model. Experiments demonstrated that integrating three single weak models produces a robust model with better performance. The E-GAE model utilizes three different models to learn the topological features and node information of graphs in parallel. Each model generates a node embedding matrix Z. The node embedding matrices generated by the three models are integrated through adaptive weighting, and the three parts of the loss produced by the models are fused using weighted coefficients. The calculation of the loss employs the binary cross-entropy function. Compared to models that process data sources independently, E-GAE integrates advanced features at different levels and expands the model’s receptive field, enhancing the ability to mine graph pattern features. This strengthens the learning capabilities and improves the model’s scalability.

Table 1 Experimental results on Cora and Citeseer datasets.

Model	GE	E-GAE	GERG	GERTA	
	Cora	
Acc	0.737	0.751	0.733	0.726	
NMI	0.528	0.540	0.526	0.519	
F1	0.714	0.727	0.720	0.707	
Precision	0.753	0.750	0.741	0.764	
ARI	0.513	0.542	0.496	0.487	
	Citeseer	
Acc	0.592	0.622	0.598	0.526	
NMI	0.304	0.349	0.309	0.271	
F1	0.546	0.580	0.576	0.490	
Precision	0.551	0.586	0.588	0.497	
ARI	0.296	0.348	0.309	0.249	

Experiment

Link prediction

Link prediction essentially constitutes a matrix-completion task, where the objective is to predict missing connections based on the presence of existing data (Liang & Gao, 2022; Wu & Cheng, 2022; Xie et al., 2023; Li et al., 2022). Three datasets were employed in the link prediction experiments: Cora, Citeseer, and PubMed. The Cora dataset comprises seven primary categories of machine learning-related papers, all of which have been cited by other papers at least once (Xie et al., 2022a). Each paper in this dataset is represented as a binary vector of 0 and 1 s, indicating the presence or absence of specific words. The dictionary contains a total of 1,433 features, and these features are binary as well (Xie et al., 2022a). The Citeseer dataset encompasses papers across seven distinct topics, each of which has been cited by other papers at least once. The node features in this dataset are also binary, indicating the presence or absence of specific papers. The PubMed dataset includes 19,717 web citations distributed across three categories and is linked by 44,338 edges. Detailed information about this dataset is listed in Table 2.

Table 2 Three datasets.

Datasets	Nodes	Edges	Features	Labels	
Cora	2,708	5,429	1,433	7	
Citeseer	3,327	4,732	3,703	6	
PubMed	19,717	44,338	500	3	

Metrics

The same evaluation metrics as those used by Pan et al. (2019), Huang & Frederking (2019), and Wou et al. (2021) were utilized to make the experimental results more convincing. The area under the curve (AUC) and average precision (AP) metrics were employed for the link prediction task. AUC represents the probability that the predicted value of the positive sample is greater than the predicted value of the negative sample for a pair of randomly drawn positive and negative samples from the sample, and a larger value indicates better model performance (Lin Li & Liu, 2021). AP is the area below the precision-recall (PR) curve, and a larger value indicates better model performance (Lin Li & Liu, 2021). This study divided the dataset into training, validation, and test sets. The test set contains 10% of the data, and the validation set contains 5% of the data (Pan et al., 2019; Wou et al., 2021).

Baseline

The baseline models were roughly divided into two categories: traditional methods, including DeepWalk, K-means, Spectral, GraphEncoder, DNGR, RTM, RMSC, TADW, and GCN-based methods, such as GAE, ARGA, RWR-GAE, and ARWR-GE.

Among the traditional methods, DeepWalk generates a sequence of nodes by performing random walks in the network, a process that may overlook some critical nodes. K-means results are highly dependent on the initial choice of centroids and require the number of clusters to be known or estimated in advance, which is unrealistic. Spectral is a commonly used efficient embedding algorithm. GraphEncoder is prone to overfitting when dealing with small datasets, leading to poor generalization ability. DNGR is a graph embedding model that uses graph structure to generate vector representations of nodes. RTM is a model for network structure and node-level relationships. RMSC is a multi-view clustering algorithm. TADW demonstrated that DeepWalk is equivalent to matrix factorization algorithms. GAE simplifies the original graph in GCN-based methods into a low-dimensional embedding during the encoding phase, which may lead to losing some critical structural information and is very sensitive to hyperparameters. ARGA introduces an adversarial training mechanism to enhance the robustness of the model, but this increases the complexity of model training and may lead to unstable training, also consuming more computational resources. RWR-GAE utilizes directed random walks to enhance the strength of relationships between nodes, but if the RWR-GAE parameters are not correctly chosen, it can lead to information distortion or increased noise. ARWR-GE essentially fuses the ARGA model with the RWR-GAE model to address the embedding problem.

Experiment parameters

The neural processes for graph neural network model uses 32 neurons for both layers in the encoder. The two multilayer perceptron (MLP) layers use 64 and 32 neurons, respectively. The initial learning rate is α = 0.01, β = [0.9, 0.009], and dropout = 0.6. The number of iterations on the Cora and Citeseer datasets is epoch = 500, and the number of iterations on the PubMed dataset is epoch = 4000. The trade-off coefficients of the EGAE model on the Cora and Citeseer datasets are 10−3, with trained iterations epoch = 200, dropout = 0.6, learning rate α = 10−3, and the encoder consisting of a hidden layer of 256 neurons and an embedding layer of 128 neurons. The DGAE model was initially trained with a learning rate of α = 0.01, iterations epoch = 200, and dropout = 0.6 on the PubMed dataset, and the encoder consists of a hidden layer of 32 neurons and an embedding layer of 16 neurons. The settings in the corresponding papers were maintained for the other baseline models. An E-GAE model is proposed, which ensembles GATE, RWR-GAE, and EGSRWR-GAE through adaptive weights to obtain a new node embedding matrix. The number of training epochs on the Cora data is 200, learning rate le = 0.01, step size l = 50, window size w = 30, and number of walks c = 50. The parameters of the Citeseer dataset remain the same as those of the Cora dataset. The total number of training epochs is 2,000 on the PubMed dataset, learning rate le = 0.1, and step size l = 70. The relevant parameters of the specific model are listed in Table 3. The SuperGAT network model has a head count of 12, the attention type used is MX, and the sampling rate of the edges is 0.8. The graph attention autoencoder uses a multi-head attention mechanism with a headcount of 3.

Table 3 Parameters on different datasets.

Parameters	Cora	Citeseer	PubMed	
Units of hidden	32–16	32–16	32–16	
Epoch	200	200	2,000	
Walk length	50	50	70	
Window size	30	30	80	
Number of walks	50	50	30	
Dropout	0.6	0.6	0.6	

Experiment results

In the experiments, the performance of the E-GAE model for link prediction was evaluated against other models on three datasets. The results of these comparisons are detailed in Table 4. Compared to the state-of-the-art NPGNN model, the E-GAE model outperformed in AP and AUC values. The E-GAE model achieved an improvement of 0.8% in AP on the Cora dataset, 1.3% in AUC on the Citeseer dataset, and 2.0% in AUC on the PubMed dataset. These improvements demonstrate the model’s advantages in the context of link prediction. The key factor contributing to this performance gain is the ensemble approach, where different models are separately trained and then combined with adaptive weights to generate a new node embedding matrix. Compared to existing models that rely on a single technique for node embedding, this approach integrates three distinct models. This circumvents the limitations of using a single model and leads to enhanced overall model performance.

Table 4 Comparison of models on link prediction tasks.

Item	Model	Cora	Citeseer	PubMed	
		AUC	AP	AUC	AP	AUC	AP	
Traditional methods	SC (Pan et al., 2019)	84.6	88.5	80.5	85.0	84.2	87.8	
DeepWalk (Pan et al., 2019)	83.1	85.0	80.5	83.6	84.4	84.1	
GCN methods	GAE (Pan et al., 2019)	91.0	92.0	89.5	89.9	96.4	96.5	
ARGA (Wou et al., 2021)	92.3	93.1	91.7	93.0	96.5	96.9	
RWR-GAE (Xie et al., 2023)	92.6	92.2	92.1	91.5	96.2	96.2	
NPGNN (Xie et al., 2023)	93.1	93.9	94.0	95.0	95.2	95.0	
Attention methods	E-GAE	93.9	94.5	92.4	93.7	96.5	97.0	

Clustering

Clustering is a well-established technique in unsupervised learning, and recent research has focused on using graph auto-encoders for clustering tasks (Wang et al., 2019). However, existing graph auto-encoders often struggle to achieve high accuracy in clustering tasks. Therefore, the E-GAE model was introduced to enhance clustering performance. The specific dataset utilized in this study is listed in Table 2.

Metrics

Similarly, the same evaluation metrics for the clustering task were used as in Pan et al. (2019), Huang & Frederking (2019), and Wou et al. (2021). This study employed Acc, NMI, ARI, F1, and Precision. Acc is accuracy, and NMI is the degree of similarity of the clusters, usually by comparing the clustering results with the true labels. The larger the value, the more similar the clustering results are (Lin Li & Liu, 2021). ARI represents the degree of similarity between the clustering results and the true classes (Lin Li & Liu, 2021). Precision indicates the proportion of samples identified as positive classes by the truly positive model, and F1 represents the summed average of the precision and recall (Lin Li & Liu, 2021).

Experiment results

Five key evaluation metrics were employed in the clustering task evaluation to assess the performance of the proposed E-GAE model. Although existing graph auto-encoders typically utilize GCN as an encoder and inner product decoding (Kipf & Welling, 2016; Weng, Zhang & Dou, 2020; Xie et al., 2023), the E-GAE model takes a different approach by integrating three graph auto-encoders. This approach resulted in improved performance in clustering tasks. Experiments on the three datasets, Cora, Citeseer, and PubMed, demonstrated the effectiveness of the E-GAE model, as detailed in Tables 5–7. Compared to state-of-the-art models, the model achieved significant improvements in various metrics. For instance, on the Cora dataset, the model showed a 9.4% increase in ARI, a 2.9% increase in NMI, and a 5.8% increase in Acc. On the Citeseer dataset, the model improved Acc by 3.9% and ARI by 4.0%. The model enhanced NMI by 2.7% on the PubMed dataset and ARI by 2.9%. In the experiments, the parameter dropout significantly impacts clustering tasks. Hence, comparative ablation experiments were conducted on the clustering task for the dropout parameter, which showed relatively good results when dropout = 0.6 on the Cora and Citeseer datasets. The specific information is shown in Table 8.

Table 5 Model comparison of clustering tasks on the Cora dataset.

Item	Model	Acc	NMI	F1	Precision	ARI	
Traditional methods	DeepWalk (Wang et al., 2019)	0.484	0.327	0.392	0.361	0.243	
k-means (Wang et al., 2019)	0.492	0.321	0.368	0.369	0.230	
Spectral (Pan et al., 2019)	0.367	0.127	0.318	0.193	0.031	
GraphEncoder (Pan et al., 2019)	0.325	0.109	0.298	0.182	0.006	
DNGR (Pan et al., 2019)	0.419	0.318	0.340	0.266	0.142	
RTM (Pan et al., 2019)	0.440	0.230	0.307	0.332	0.169	
RMSC (Pan et al., 2019)	0.407	0.255	0.331	0.227	0.090	
TADW (Pan et al., 2019)	0.560	0.441	0.481	0.396	0.332	
GCN methods	GAE (Pan et al., 2019)	0.594	0.426	0.591	0.593	0.343	
ARGA (Wou et al., 2021)	0.640	0.447	0.617	0.644	0.353	
RWR-GAE (Xie et al., 2023)	0.655	0.469	0.618	0.629	0.417	
ARWR-GE (Xie et al., 2023)	0.667	0.482	0.620	0.631	0.417	
EGAE (Xie et al., 2023)	0.693	0.511	–	–	0.448	
Attention methods	E-GAE	0.751	0.540	0.727	0.750	0.542	

Table 6 Model comparison of clustering tasks on the Citeseer dataset.

Item	Model	Acc	NMI	F1	Precision	ARI	
Traditional methods	k-means (Wang et al., 2019)	0.540	0.305	0.409	0.405	0.279	
Spectral (Pan et al., 2019)	0.239	0.056	0.299	0.179	0.010	
GraphEncoder (Pan et al., 2019)	0.225	0.033	0.301	0.179	0.010	
DNGR (Pan et al., 2019)	0.326	0.180	0.300	0.200	0.044	
RTM (Pan et al., 2019)	0.451	0.239	0.342	0.349	0.203	
RMSC (Pan et al., 2019)	0.295	0.139	0.320	0.204	0.049	
TADW (Pan et al., 2019)	0.455	0.291	0.414	0.312	0.228	
GCN methods	GAE (Pan et al., 2019)	0.408	0.176	0.372	0.418	0.124	
ARGA (Wou et al., 2021)	0.572	0.348	0.544	0.572	0.340	
RWR-GAE (Xie et al., 2023)	0.559	0.312	0.527	0.545	0.290	
EGAE (Xie et al., 2023)	0.583	0.334	–	–	0.308	
Attention methods	E-GAE	0.622	0.349	0.580	0.586	0.348	

Table 7 Model comparison of clustering tasks on the PubMed dataset.

Item	Model	Acc	NMI	F1	ARI	
Traditional methods	DeepWalk (Wang et al., 2019)	0.543	0.102	0.530	0.088	
k-means (Pan et al., 2019)	0.580	0.278	0.544	0.246	
Spectral (Pan et al., 2019)	0.496	0.147	0.299	0.010	
GraphEncoder (Pan et al., 2019)	0.531	0.210	0.506	0.184	
DNGR (Pan et al., 2019)	0.468	0.153	0.445	0.059	
M-NMF (Pan et al., 2019)	0.470	0.084	0.443	0.058	
RMSC (Pan et al., 2019)	0.629	0.273	0.521	0.247	
TADW (Pan et al., 2019)	0.565	0.224	0.481	0.228	
GCN methods	GAE (Pan et al., 2019)	0.632	0.249	0.511	0.246	
DGAE (Xie et al., 2023)	0.684	0.290	–	0.291	
Attention methods	E-GAE	0.700	0.317	0.690	0.320	

Table 8 Contrast ablation experiments on clustering.

	Cora	
Dropout	0.5	0.6	0.7	
Acc	0.726	0.751	0.732	
NMI	0.532	0.540	0.523	
F1	0.705	0.727	0.712	
Precision	0.742	0.750	0.763	
ARI	0.508	0.542	0.493	
	Citeseer	
Dropout	0.5	0.6	0.7	
Acc	0.604	0.622	0.595	
NMI	0.331	0.349	0.331	
F1	0.576	0.580	0.568	
Precision	0.585	0.586	0.581	
ARI	0.326	0.348	0.327	

These results demonstrate that the E-GAE model outperforms most existing models across multiple evaluation metrics. Clustering was performed on the Cora and Citeseer datasets to provide a visual representation of the model’s effectiveness, and t-distributed stochastic neighbor embedding (t-SNE) was employed to visualize the results in two dimensions, as illustrated in Fig. 3.

Figure 3 Cora and Citeseer datasets model visualization.

Time comparison

In this approach, the previous single models were not used; instead, an integrated method was adopted to learn the embedded representations of nodes. However, this integrated approach can make the model more computationally intensive. Therefore, the RWR-GAE and E-GAE models were compared in terms of running time on the Cora, Citeseer, and PubMed datasets. The average running times over ten epochs for each model on each dataset were calculated. The experiments revealed that the RWR-GAE model is significantly faster than the E-GAE model. Specifically, the RWR-GAE model is still essentially GAE, using only one GCN as an encoder. The difference is that the E-GAE proposed in this study is an integration model, essentially a variant of GAE. It uses three encoders to learn the feature representation of nodes in parallel and finally fuses the learned features. Therefore, the computational time and spatial complexity of the RWR-GAE model are smaller than those of the E-GAE model proposed in this study. From the experimental results, the average calculation time of the E-GAE model is about three times that of the RWR-GAE model every ten epochs, which is consistent with the conjecture. However, the E-GAE model outperforms the RWR-GAE model in terms of clustering and link prediction. Therefore, the choice of model should be based on specific needs and priorities. For a detailed comparison of the running times, please refer to Table 9. Based on the experimental results in Table 9, the model still has an advantage over traditional methods. Taking the RWR-GAE model as an example, although the time cost of this method is relatively low, its performance is not satisfactory, and it even lags behind the ARGA model. The proposed method achieves a comprehensive lead in multiple evaluation metrics on multiple datasets and takes an acceptable amount of time. Compared to the general method of sacrificing running time to improve performance, this method is more comprehensive.

Table 9 Model time comparison (seconds).

Model	Cora	Citeseer	PubMed	
AGC (Kang et al., 2022)	3.42	40.36	20.77	
DAEGCC (Kang et al., 2022)	561.69	946.89	50,854.15	
FGC (Kang et al., 2022)	4.60	9.49	268.44	
RWR-GAE (our)	0.57	0.85	–	
E-GAE	2.57	3.10	37.31	

Discussion

The proposed E-GAE model adopts a distinct approach compared to traditional graph embedding models. Although traditional models can use only node features or graph structure to learn feature representations of nodes, the E-GAE model employs an integrated approach incorporating node features and topological information from the graph. This results in a more complex model but allows for a better feature representation of nodes. Other models, such as the ARWR-GE model proposed by Wou et al. (2021), also aim to address issues with existing models that ignore the potential data distribution in the graph and fail to capture contextual information. However, the proposed approach does not consider the data distribution problem, as this method is not the most optimal solution. In the context of large-scale graphs, traditional graph auto-encoder models do not effectively capture valuable information. Jiao et al. (2023) proposed a self-supervised method for comparing subgraphs, which can better represent node information. Although the proposed model has demonstrated good results, it has a higher time complexity on large graphs and takes longer to run. Hence, the use of subgraphs can be considered as an alternative approach. Maintaining high performance on large graphs while reducing computational demands by incorporating subgraphs into the modeling process is possible. In addition, the E-GAE model uses a binary cross-entropy loss function, but it tends to bias toward the majority class when the sample categories are not balanced, causing poor classification results. The E-GAE model has high requirements for the balance of sample classification in the dataset and thus has certain limitations.

Conclusion

Traditional models typically use a single encoder to learn the embedding representation of nodes, which has limitations. The E-GAE model utilizes adaptive weights to integrate three graph auto-encoders, resulting in new node embedding representations. This integrated approach effectively addresses issues where reconstruction loss ignores node representation and fails to capture the contextual information of nodes within the graph data, leading to lower-quality embeddings. The E-GAE model is different from the previous model as it uses the integration idea. The GAE model is not limited to integrating RWR, EGSRWR-GAE, and GATE models and can also be replaced with other models. The experiments demonstrated that, in clustering tasks, the effect of the integrated approach on other models still has advantages, proving the program’s feasibility. Although the proposed E-GAE model demonstrates superiority over other baseline models in node clustering and link prediction tasks, it has a high time complexity and memory usage when dealing with large-scale graph data, significantly reducing computational speed. In addition, the presence of many isolated nodes in the graph can also affect the model’s performance. Future research can explore the number of integrated models and the effect of different models on link prediction and clustering tasks, as well as investigate ways to reduce memory usage and accelerate computational speed. Meanwhile, we will persist in exploring the extension of the E-GAE model to the biomedical field for the prediction of related diseases, such as Alzheimer’s disease

Supplemental Information

Supplemental Information 1 Code.

Additional Information and Declarations

Competing Interests

The authors have no conflicts of interest to disclose.

Author Contributions

Chengxin Xie conceived and designed the experiments, performed the experiments, performed the computation work, authored or reviewed drafts of the article, and approved the final draft.

Jingui Huang conceived and designed the experiments, analyzed the data, performed the computation work, authored or reviewed drafts of the article, and approved the final draft.

Yongjiang Shi analyzed the data, prepared figures and/or tables, and approved the final draft.

Hui Pang performed the experiments, performed the computation work, prepared figures and/or tables, authored or reviewed drafts of the article, and approved the final draft.

Liting Gao performed the experiments, prepared figures and/or tables, and approved the final draft.

Xiumei Wen conceived and designed the experiments, performed the computation work, authored or reviewed drafts of the article, and approved the final draft.

Data Availability

The following information was supplied regarding data availability:

The code is available at GitHub and Zenodo:

- https://github.com/xcgydfjjjderg/graphautoencoder

- Chengxin, X. (2024). Ensemble Graph auto-encoder. Zenodo. https://doi.org/10.5281/zenodo.12621055.

The code contains the Cora, Citeseer, and PubMed datasets used in this article.

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
