# Peer review of "Ensemble graph auto-encoders for clustering and link prediction"

_PeerJ Computer Science, doi:10.7717/peerj-cs.2648_

## Round 0.1 · original submission · Major Revisions

Dear Authors,

I suggest you go through all the comments provided by reviewers (especially 1,3 and 4) and carefully clarify ALL their doubts.

Moreover, It is still not clear the novelty of your research and your contribution.

Please, address all these points and then resubmit.

Best regards

M.P.

Reviewer 1 ·

Basic reporting

In this paper, authors proposed a novel ensemble graph embedding technique, called as: E-GAE which is a combination between graph auto-encoding through random walk and matrix ensemble strategy to learn the representations of input graph in context of unsupervised/contrastive learning. Specifically, within this paper, in the first step, authors have utilized different graph neural network (GNN)-based architectures to achieve the multi-viewed representations of the input graph. Then, the different views of graph representations are combined to produce the unified network node embedding matrix, which are later utilized to reconstruct the original graph structure through the auto-encoding learning paradigm – to achieve the reconstruction loss/optimization learning strategy for the proposed technique authors have combined the random walk strategy with previous skip-gram architecture. Generally speaking, the proposed ideas within this paper can be considered as novel and interesting in which the random walk strategy is apply to facilitate the process of graph reconstruction, improving the local contextual node structure focusing as well as reduce the time-complexity for the optimization process. Authors also provide sufficient empirical studies within benchmark networked datasets to demonstrate for the effectiveness/outperformance of authors’ proposed techniques in this paper. Beside positive comments of authors’ work this paper, I also have some revision suggestions/queries for authors to improve their paper quality, including:
1) First of all, the structure and presentation of this paper are really problematic as the illustrative figures, charts, tables, etc. are all put outside the main contents of the paper, thus it is quite difficulty for the reader to refer the contents in these figures/charts/tables. Therefore, I suggested authors to carefully check on the current presentation of their paper.
2) Moreover, the contents of the introduction section should be also carefully revised in order to make it clearer on the main contributions of authors in this paper, what are their novelty, original contributions which make their studies are different from previous graph embedding technique?
3) More discussions about drawbacks of previous techniques within the related works section as well as how authors’ proposed technique within this paper can assist to overcome these limitations?
4) More explanation as well as illustrative figures, including step-by-step pseudo-code like illustrations, etc. – paragraph starting “Random walk Graph Auto-Encoder (RWR-GAE)” - should be added to clearly describe, as currently it is quite confusing to understand.

5) For the graph reconstruction process, please also provide the formulation of the general loss function which is applied for the model optimization strategy.
6) Finally, authors also should provide the extensive discussions about the computational time/space complexity analysis of the proposed model within this paper.

Experimental design

No comment.

Validity of the findings

Please refer to my basic reporting section.

Additional comments

No comment.

Cite this review as

Reviewer 2 ·

Basic reporting

no comment

Experimental design

no comment

Validity of the findings

no comment

Cite this review as

·

Basic reporting

This paper introduces the E-GAE model for graph auto-encoders, combining integration techniques to improve graph embedding tasks. Below are detailed technical comments that you can use for a paper review:

- Clarity of Contribution: The introduction provides a good overview of the importance of graph auto-encoders, but the specific novelty of the E-GAE model needs more emphasis. The paper should clarify how E-GAE differentiates itself from existing methods in graph embeddings, particularly in link prediction and clustering tasks.
- Research Gap: While the introduction touches on the limitations of previous models, it could benefit from a more explicit statement of the research gap. Clearly outlining the shortcomings of prior models and how E-GAE addresses these would help to strengthen the motivation for this research. Also add in-depth analysis of this in the related work section.

- Clarity and Informativeness: While the figures and tables present the results clearly, they could benefit from more informative labels and captions. For example, graphs that compare the performance of different models should highlight key trends or outliers to make the comparisons easier to interpret.
- Grammar and Style: Please double-check the grammar errors and typos in the paper. The authors are requested to check the whole manuscript via a professional English proofreading service to ensure the article is free of language issues. The authors need to attach the proofreading certificate when submitting the revised manuscript.

Experimental design

- Comparative Framework: The paper could benefit from a clearer comparison framework that organizes the existing methods by category (e.g., GCN-based methods, attention-based methods) and directly contrasts their weaknesses with the strengths of E-GAE.
- Model Architecture: The methodology section explains the E-GAE architecture, but more detailed diagrams could clarify the process flow and key components. Specifically, a step-by-step breakdown of how the integration process works and its advantages in feature extraction and error minimization would be helpful.
- Integration Strategy: The paper mentions the integration of different features and topological information, but the exact integration strategy needs to be more thoroughly explained. For instance, how does E-GAE merge different information streams, and what advantage does this approach provide over models that process these data sources independently?
- Computational Efficiency: The paper should include more discussion on the computational complexity of E-GAE, especially given its integrated structure. An analysis of time complexity, memory usage, and scalability to larger graphs would provide valuable insights into its practicality for real-world applications.
- Dataset Choices: The paper uses three publicly available datasets to test the E-GAE model, but the reasoning behind the selection of these datasets needs more elaboration. How do these datasets capture the diversity or complexity of typical graph embedding tasks? Justifying their relevance would enhance the robustness of the experiment.
- Evaluation Metrics: The paper focuses on link prediction and clustering accuracy as key metrics, which is appropriate. However, including additional performance metrics such as computational time, memory consumption, or precision/recall for link prediction could provide a more comprehensive evaluation of the model.
- Baseline Comparisons: The baseline methods used for comparison should be discussed in more detail. Are these baseline models representative of the current state of the art? It would also be beneficial to include more competitive models for comparison to demonstrate the efficacy of E-GAE under a wider variety of conditions.
- Limitations and Trade-offs: The discussion briefly mentions potential future improvements but does not address the current model's limitations. What are the trade-offs of using E-GAE, such as increased computational complexity or sensitivity to hyperparameters? Addressing these would provide a balanced view of the model's performance.

Validity of the findings

- Performance Discussion: The results section demonstrates the superiority of E-GAE in terms of link prediction and clustering accuracy, but the discussion could be strengthened by providing a more nuanced analysis of the results. For instance, in which cases does E-GAE excel, and are there specific types of graphs or conditions where its performance might degrade?

Additional comments

none

Cite this review as

·

Basic reporting

English is generally clear and unambiguous

The bibliographic references seem close to the theme of the paper and are sufficiently provided.

The structure of the article is disturbing: the figures and tables are in appendices which leads the reviewer to go back and forth to see them. In addition, there are many reminders of definitions of models and techniques which are useless

The results displayed support the hypotheses made at the beginning

The poverty of the paper comes from the fact that the formal results do not include too many clear definitions of all the terms and theorems, as well as detailed proofs.

Experimental design

The proposed research is not very original, it takes up a lot of what already exists and makes new associations. Moreover, even this point is not very invested: there is a lack of justification

The research problem is well defined, relevant and significant. On the other hand, it is not well indicated how the research fills an identified knowledge gap.

The investigation lacks rigor, there is a lack of formal description of the methods used.

The methods are not described with sufficient detail and information to be reproduced

Validity of the findings

The impact and novelty are seriously evaluated using 3 data sets

All the data sets used for the evaluation are described. They have been published and used by all the researchers in the field. The only regret is that there are no data examples to show where the method studied failed

The conclusions are well formulated but we would have liked to have a paragraph on the perspectives of this research, showing new avenues to explore

Additional comments

Detailed remarks and questions
• The first part of subsection 3.1 up to GCN is useless, it repeats the same remarks seen in the introduction. Similarly, the beginning of subsection 3.2 on the definition of GAE is unnecessary. It can be moved to the introduction or Related works
• Section 3.1, which is supposed to explain the nature of the proposed model, talks about the model of Kipf et al. which was already explained in Related works. Similarly for section 3.2. What is missing is your positioning in relation to these models
• It was necessary to wait until line 251 to know which model you are going to use, which is too late
• In equation (1), there is a missing closing parenthesis
• The references to the equations, for example (2) and (3) should be removed and put in the surrounding text
• In subsection 3.2 you either show the interest of using GAEs to solve the problems mentioned in the introduction, or you remove it because it does not bring anything new.
• It is difficult to understand the interest of figure 1. It must either be described better or removed.
• In Table 4, it is better to put in bold only the best scores
• What do the 3 columns in Table 9 represent?
• In Tables 4, 5, 6, 7, 8 and 9, you must put the references of the articles next to the names of the models used

Cite this review as

---

## Round 0.2 · Minor Revisions

Dear Authors,

In order to improve the clarity and the readability please take in to account the issue raised by Reviewer #3.

Best,

Reviewer 1 ·

Basic reporting

After carefully checking all revisions and authors’ responses, I confirmed that authors have completed fulfilled all my and other reviewers’ recommendations, all problems of previous paper have been resolved, thus I thought this paper can be accepted for publication in this form. Thanks.

Experimental design

No comment.

Validity of the findings

Please refer to my basic reporting section.

Additional comments

No comment.

Cite this review as

·

Basic reporting

The authors are requested to have the entire manuscript checked by a professional English proofreading service to ensure that the article is free of language issues.

Experimental design

no comment

Validity of the findings

no comment

Additional comments

The authors have addressed all of my comments. The only issues left are the language issues in the paper.

Cite this review as

·

Basic reporting

The authors attempt through this work to address the problems of graph models that ignore the data distribution and fail to capture contextual information. The authors propose the E-GAE model that uses an embedded approach that integrates both node features and graph topological information. The proposed E-GAE is integrated from three graph autoencoders, but other integrations are also compared.

English is generally clear and unambiguous. The bibliographic references seem close to the theme of the paper and are sufficiently provided

The revision of the paper allowed the authors to clarify several points and make the article both readable and very rich in new concepts.

Experimental design

The impact and novelty are seriously evaluated using 3 data sets. All the data sets used for the evaluation are described. They have been published and used by all the researchers in the field. The only regret is that there are no data examples to show where the method studied failed.
The conclusions are well formulated but we would have liked to have a paragraph on the perspectives of this research, showing new avenues to explore.
The proposed research is very original, it takes up a lot of what already exists and makes new associations. The research problem is well defined, relevant and significant.

Validity of the findings

The article contains many new findings that will impact research in this area. .

All underlying data have been provided; they are robust, statistically reliable, and controlled.

The conclusions are well formulated, linked to the original research question, and limited to the supporting results.

Additional comments

The article is well written. It could be better worded and illustrated but it contains enough new ideas that are well expressed.

Cite this review as

---

## Round 0.3 · Minor Revisions

Dear Authors,

you have addressed all the comments raised by the reviewers.

Anyway, reading again the manuscript I did not find any link or mention to repositories for your code. I think that for addressing transparency and reproducibility in science you should provide such a link.

Please, share your code on a public repository (e.g. Github) and put the link on the paper.


M.P.

·

Basic reporting

The authors have addressed all comments. This paper is now recommended for publication.

Experimental design

The authors have addressed all comments. This paper is now recommended for publication.

Validity of the findings

The authors have addressed all comments. This paper is now recommended for publication.

Additional comments

The authors have addressed all comments. This paper is now recommended for publication.

Cite this review as

---

## Round 0.4 · accepted · Accept

Dear Authors,

You addressed all the points raised by the reviewers and the editor.
Your manuscript is ready for publication.

Best regards,

M.P.